# Ricci-GNN: Defending Against Structural Attacks Through a Geometric Approach

## Abstract

Graph neural networks (GNNs) rely heavily on the underlying graph topology and thus can be vulnerable to malicious attacks targeting at perturbing graph structures. We propose a novel GNN defense algorithm against such attacks. In particular, we use a robust representation of the input graph based on the theory of graph Ricci flow, which captures the intrinsic geometry of graphs and is robust to structural perturbation. We propose an algorithm to train GNNs using re-sampled graphs based on such geometric representation. We show that this method substantially improves the robustness against various adversarial structural attacks, achieving state-of-the-art performance on both synthetic and real-world datasets.

## 1 Introduction

Recent years we have witnessed the success of graph neural networks (GNNs) on many graph applications including graph classification (Xu et al., 2019b), node classification (Kipf & Welling, 2016; Veličković et al., 2018), graph generation (You et al., 2018) and recommendations (Ying et al., 2018). As GNNs have shown great potentials, their vulnerability to adversarial attacks (Szegedy et al., 2014; Goodfellow et al., 2015) becomes a serious concern that hinders their deployment in real life critical applications. For example, a GNN algorithm for fraud detection in financial transaction graphs (Wang et al., 2019a) needs to be robust against attacks aiming at disguising fraud transactions as normal ones. In health informatics, prediction of polypharmacy side effects (Zitnik et al., 2018) must be robust against attacks that intend to endanger certain patients. In a recommendation system, the developers need to consider potential attacks from spammers who may create fake followers to increase the influence scope of fake news (Zhou & Zafarani, 2018).

One way to attack a GNN model is to modify the graph topology by inserting or deleting edges (Jin et al., 2020a). A small perturbation of the network topology can significantly impair the graph neural network's performance (Dai et al., 2018; Zügner & Günnemann, 2019b). For example, Meta-Attack (Zügner & Günnemann, 2019a) can increase the misclassification rate of GCN on a political blog data set by over 18% with only 5% perturbed edges. This is not surprising as graph topology is essential for GNNs, both as the backbone of a GNN architecture and as important structural features. In particular, the local neighborhood of each node is commonly used to define receptive fields for the convolution operator. The statistics of local neighborhood, e.g., node degrees, are important structural information used as additional node features (Veličković et al., 2018) to re-calibrate the convolutional operation (Kipf & Welling, 2016).

In this paper, we focus on defending against global poisoning adversarial attacks which corrupt the graph topology in the training phase. Some existing approaches assume the graph is true and leverage known robust training techniques, e.g., enforcing priors on latent representation of data (Zhu et al., 2019). These solutions can still be limited by the corrupted graph, considering how critical the underlying graph is for a GNN model. Other methods assume prior knowledge on the graph topology, and perform graph restructuring, e.g., via low-rank filtering (Entezari et al., 2020) or graph specification (Wu et al., 2019), hoping to remove abnormal edges from the attack. These strong priors, although proven useful, also limit the generality of the method.

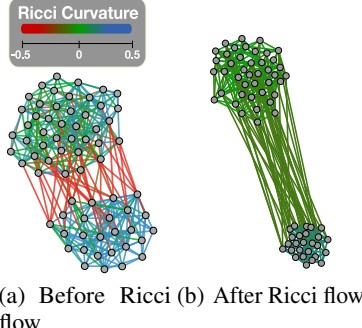

(a) Before Ricci (b) After Ricci flow
flow

Figure 1: An illustrative example of Ricci curvature and Ricci flow on graphs. 1(a): The bridge edges (red) between communities have negative curvature while the edges inside communities (blue) have positive. 1(b): The same graph after Ricci flow, in which length of edges are proportional to weights (Ricci flow metric). Nodes within one community are moved closer whereas the two communities are moved further apart.

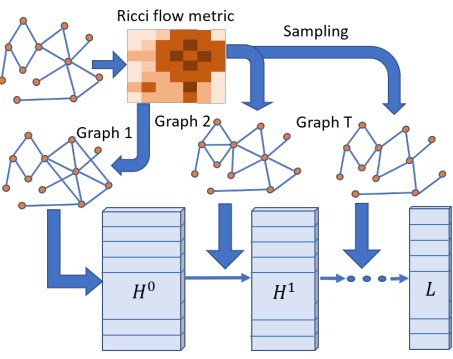

Figure 2: An overview of our Ricci-GNN. We first compute the Ricci flow metric from the input (attacked) graph and re-sample edges by using a Gaussian filter on each node. A newly sampled graph is used for the training phase in each iteration of a standard GCN.

## 1.1 A GEOMETRIC VIEW OF GRAPHS

We take a novel direction to find a *robust representation* of the graph topology through a geometric lens. We view a discrete graph in a continuous framework, in which nodes stay in an underlying metric space and the connectivity of two nodes has a stochastic nature, depending on the features of the two nodes, their respective neighborhoods and the entire node distribution. The input graph $G$ is replaced by an *ensemble of graphs*, considered as (randomized) discrete realizations of the same underlying metric space in which $G$ is taken. In order to do that, we recover the *metric distance* between two nodes in the underlying space through the Ricci flow metric on the input graph $G$. Note that we are not trying to explicitly find an embedding which would involve choices (e.g, Euclidean vs non-Euclidean, dimensionalities) that introduce extra and unnecessary distortion. Instead, we represent the underlying metric space via pairwise geodesic distance between nodes.

Our geometrical approach is inspired by the Riemmanian geometry in the continuous setting (Hamilton, 1982; Perelman, 2002). On a Riemmanian manifold, one can define Ricci curvature to measure the amount of 'bending' or 'curving' at each point. With Ricci curvature, one can define a diffusion process by changing the Riemannian metric (stretching or shrinking locally) such that curvature is uniform everywhere. This uniformization process is called Ricci flow. This theory can be extended to a graph setting (Ollivier, 2009). Generally speaking, edges that are locally well connected have positive curvature while edges that are locally sparsely connected have negative curvature. In Ricci flow, edges of negative curvature are stretched (with increased edge weight) and edges of positive curvature are condensed (with decreased edge weight). These new edge weights that uniformize the Ricci curvature of the graph are called the Ricci flow metric. See Figure 1 for an illustration. Graph Ricci curvature and Ricci flow can be used to identify critical edges in a graph (Ni et al., 2015; Sandhu et al., 2015) and to identify community structures (Ni et al., 2019; Sia et al., 2019). We also note that graph Ricci curvature has been used in GNN for node classification task (Ye et al., 2020), but not for defending structural attacks to GNN.

**Robustness against topological perturbation.** Ricci flow metric has been shown to be robust to random deletion and addition of edges (Ni et al., 2018). This attributes to the fact that Ricci flow is a global process that tries to uncover the underlying metric space supported by the graph topology and thus embraces redundancy. Compared to other graph metrics such as the hop count metric and metric obtained by spectral embedding, Ricci flow metric provides a better trade-off between robustness and representation power of the graph metric, as shown in Figure 3. When two edges are deleted, the Ricci flow metric is rarely affected (Figure 3 (a)), similar to the hop count metric (Figure 3 (c)); while the distance metric by spectral embedding is substantially more sensitive (Figure 3 (b)). We note that the hop count metric is also robust to dynamic edge deletions due to the small world phenomena and multiple shortest paths in the graph; however the hop count metric takes only integer values and generally lacks descriptive power to provide desirable resolution and differentiation.

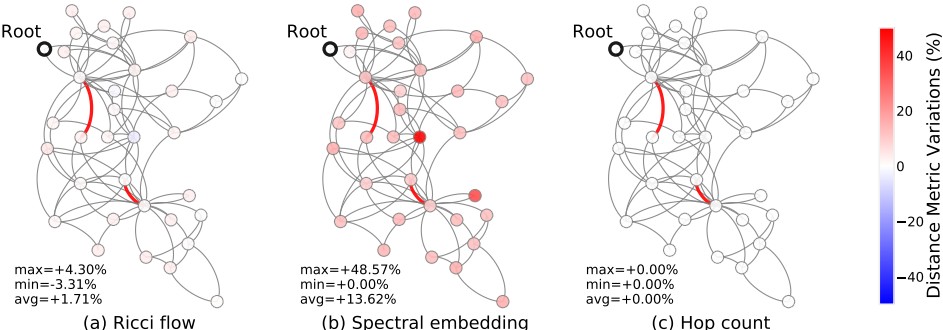

Figure 3: Changes in distance between all nodes and a fixed root in karate club graph for (a) Ricci flow metric, (b) spectral embedding, and (c) hop count, when two edges (shown in red) are removed from the network. Vertices are colored proportional to the magnitude of variations in distance.

To train a GNN using the Ricci flow metric, we generate an ensemble of sample graphs $G_1, G_2, \cdots$, and use a new sample in each network layer of the GNN of every training epoch (Figure 2). Therefore the trained model is enforced to focus on the underlying metric information represented by the graph (which is much more robust) and not on the particular input graph topology (which could be corrupted) per se.

Our method is agnostic to both models and attacks, thus can be applied to different GNNs and different structural attacks. We show in both synthetic and real-world datasets that the proposed algorithm effectively defends against various structural attacks, with improved performance compared to other defense schemes.

We summarize our contributions as follows.

- We are the first to take a geometric view of the GNN defense problem. We propose to train GNNs with the Ricci flow representation of a graph instead of its attacked topology.

- We design a new algorithm to sample graphs based on the Ricci flow representation for training GNN. This effectively alleviate the impact of structural attacks by adversaries.

- We demonstrate the efficacy of our method on various synthetic and real-world datasets, against state-of-the-art graph topology poisoning methods.

## 1.2 RELATED WORK

The vulnerability of deep neural network models w.r.t. adversarial attacks is well known. And graph neural networks are not an exception (Dai et al., 2018; Zügner et al., 2018; Zügner & Günnemann, 2019a). Here we briefly review the methods for attacking and defending against GNNs.

**Adversarial attack on graphs.** There are two categories of attacks: evasion attacks and poisoning attacks. Evasion attacks generate fake samples for the trained model in the testing time, while poisoning attacks directly modify the training data. Dai et al. (2018) employs a reinforcement learning based framework for non-targeted test-time attacks (i.e. evasion) on graph classification and node classification. The focus is on the modifications of graph structures, and the attackers are restricted to edge deletions only. Zügner et al. (2018) consider both training-time (i.e. poisoning) and testing-time attacks. The attacks, called nettack, are based on a surrogate model with both edge insertion and deletion. Nettack is a local attack, where the goal is to lower the performance on a target node. Later, a *meta-learning* poisoning attack is developed by Zügner & Günnemann (2019a) which aims to decrease classification accuracy globally. It treats the the graph structure as a hyper-parameter and conducts training-time attacks through meta learning. Last, Xu et al. (2019a) proposes a gradient-based attack method that directly tackling the dicrete graph data. Since these two are the state-of-the-art non-targeted global attack method, we will mainly focus on developing defense schemes against them.

**Robustness of GNNs.** To defend against these graph attacks, Miller et al. (2019) seek to increase model robustness by decoupling structure from attributes in the classifier and re-selecting the training

data. But their method exhibits a trade-off between robustness and performance, i.e. the performance drops on clean data. Wang et al. (2019b) proposed graph encoder refining and adversarial contrastive learning. They investigate the vulnerabilities in every aggregation layer and the perceptron layer of a GNN encoder, and apply dual-stage aggregation and bottleneck perceptron to address those vulnerabilities. They mainly focus on targeted node attacks (e.g. Nettack) instead of global topology attacks. RGCN (Zhu et al., 2019) treats node features as a Gaussian distribution and encode the hidden representation of nodes by mean and variance matrices. They apply self-attention on the variance matrix to aggregate messages from neighboring nodes. However, this method only focuses on defense against random noise on node features. GCN-Jaccard (Wu et al., 2019) pre-processes the network by eliminating edges that connect nodes with sufficiently small Jaccard similarity of features. GCN-SVD (Entezari et al., 2020) proposes to vaccinate GCN with the low-rank approximation of the perturbed graph. Most of these existing methods provide insight of robustness from the perspective of optimization or matrix ranks. DropEdge (Rong et al., 2019) randomly removes a certain amount of edges from the input graph at each training epoch. It is designed to resolve the over-fitting and over-smoothing issue of developing deeep GCNs. However, it can also be used for improving the graphs robustness. Pro-GNN (Jin et al., 2020b) jointly learns a structural graph and a robust graph neural network model from the perturbed graph guided by exploring the graph properties of sparsity, low rank and feature smoothness to design robust graph neural networks. In this paper, we understand the graph robustness from a geometric view and provide an efficient sampling based model.

## 2 GEOMETRIC RE-SAMPLING FOR ROBUST GNNs

Our method uses the robust geometric representation of the input graph to train a GNN by randomly sampling a new graph based on the Ricci flow representation. Figure 2 shows the general framework. Our method is agnostic to both the GNN model and the attack strategy . We start by a brief review on graph neural networks and poisoning attacks against graphs.

### 2.1 BACKGROUND: GNNs AND POISONING ATTACKS

We focus on the semi-supervised node classification task. Consider a graph $G = (V, E)$ with node features $H = (h_1, h_2, \cdots, h_N)$, $h_i \in \mathbb{R}^D$, where $N$ is the number of nodes and $D$ is the feature dimension of each node. Only part of the graph nodes $V_\ell \subseteq V$ are labeled and our main task is to predict the label of the remaining nodes $V_u \subseteq V$ given the node features $H$, edges $E$ and labels of $V_\ell$.

A GNN essentially learns the low-dimensional representation of the nodes given the node features and graph structure. There have been various types of GNNs (Bruna et al., 2014; Gori et al., 2005), usually classified into two categories: the spectral ones and the spatial ones. Spectral graph neural networks extend CNNs to graphs by defining convolution filters in the spectral domain (Bruna et al., 2014; Defferrard et al., 2016; Kipf & Welling, 2016). They utilize the concept of graph Fourier transformation, and define the spectral filters on the eigenvalues of the graph Laplacian matrix. Spatial graph neural networks, on the other hand, define graph filters in the spatial domain. They iteratively update graph nodes' representation by aggregating information from neighbors (Gilmer et al., 2017). We can see both Laplacian matrix and the neighborhood relationship essentially represent the graph structure information, thus both types of GNNs highly rely on the graph topology and are sensitive to structural perturbations.

Take the well known graph convolutional network (GCN) as an example (Kipf & Welling, 2016). GCN consists of multiple layers. Layer $t$ updates node representation from $H^{t-1}$ to $H^t$: $H^t = \sigma(\hat{A} H^{t-1} W^t)$ where $H^0 = H$, $W^t$ is the parameter for layer $t$, $\hat{A}$ is a normalized version of adjacency matrix $A$: $\hat{A} = \widetilde{D}^{-1/2} \widetilde{A} \widetilde{D}^{-1/2}$ with $\widetilde{A} = A + I$ and $\widetilde{D}$ as the degree matrix of $\widetilde{A}$.

As introduced in Section 1.2, a GNN is vulnerable to adversarial attacks. Usually the attackers introduce unnoticeable perturbations by imposing restrictions to ensure that the attack preserves the graph structure and node features. A non-targeted structural poisoning attack on graph $G$ can be formulated as the following optimization problem:

$$\arg\min_{G' \in \Phi(G)} \mathcal{L}_{\text{attack}}(f_{\theta^*}(G')) \qquad (2.1)$$

where $f_\theta$ is the GNN function for node embedding where $\theta$ is the set of parameters, $\Phi(G)$ is the constraint set for the perturbed graph, $\theta^* = \arg\min_\theta \mathcal{L}_{\text{train}}(f_\theta(G'))$. As indicated in (Zügner et al.,

2018; Zügner & Günnemann, 2019a), by treating the graph structure matrix $A$ as a parameter (or hyper-parameter) and solving this optimization problem, the attackers can significantly decrease classification performance.

## 2.2 Geometric Restructuring of Graphs

With the same insight as the *unsupervised manifold hypothesis* (Cayton, 2005; Narayanan & Mitter, 2010; Rifai et al., 2011) (real data in high dimensional spaces concentrate near low-dimensional manifolds), we view a graph as a discretization of an underlying manifold. Any manifold can be described by a collection of (local) charts, where each chart is homeomorphic to an open set in a Euclidean space and charts that overlap are compatible (transition from one chart to another is differentiable) (Lee & Lee, 2009). This will allow one to define the geodesic distance between two points on the manifold.

By using graph Ricci flow, we can recover this latent metric space that is intrinsic to the input graph, which is robust to topological perturbations. Specifically, the discrete Ollivier Ricci curvature (Ollivier, 2009) $\kappa_{xy}$ of the edge $(x, y)$ involves the ratio between the Wasserstein distance (the optimal transport distance) $W(m_x, m_y)$ and their geodesic distance $d(x, y)$

$$\kappa_{xy} = 1 - \frac{W(m_x, m_y)}{d(x, y)}. \tag{2.2}$$

where $m_x, m_y$ are two distributions defined on the neighborhood of $x$ and neighborhood of $y$, respectively. The details are in the Appendix A.1. The computation of Ricci flow metric involves multiple iterations until edge weights do not change much (Ni et al., 2015; 2019; 2018). In each iteration, we calculate the Ricci curvature of each edge, and adjust the current edge weight with a value proportional to the edge curvature. For the $t$-th iteration, all the new edge weights $w^{(t+1)}$ are calculated as

$$w^{(t+1)}(x, y) = d^{(t)}(x, y) - \kappa_{xy}^{(t)} d^{(t)}(x, y). \tag{2.3}$$

We re-normalize all the edge weights to keep the total edge weight unchanged at the end of each iteration. To speed up the computation process, we use the Sinkhorn distance (Cuturi, 2013) (on a sampled neighborhood) as an approximation of optimal transport distance. In practice, it takes less than 1 sec to compute Ricci curvature for Cora and Citeseer data set (of 4732 edges), and 6.9 secs for Polblogs (with 16714 edges) on a 36 cores machines. The detailed formulas of computing Ricci curvature and Ricci flow can be found in the Appendix A.2.

**Robustness of Ricci flow metric**    One way to visualize this is to consider embedding the graph on a manifold with uniform curvature. The edge weight on $(u, v)$ describes the proximity of $u, v$ on the manifold. In other words, it would require a lot of changes to the graph connectivity to create significant changes in this metric space and the underlying manifold. The benefit of the new metric, according to the curvature definition, is the following nice property that implies robustness to connectivity perturbation: the optimal transport distance from a distribution on neighbors of $u$ to a distribution on neighbors of $v$ is $(1 - \kappa)d(u, v)$, which is just $d(u, v)$ when curvature $\kappa$ converges to zero. In other words, there are paths connecting neighbors of $u$ and neighbors of $v$ which bypass edge $(u, v)$ and have similar length (in an average sense). This suggests that the removal of edge $(u, v)$ has small changes to lengths of (shortest) paths in the graph. This is a good thing to have, as opposed to a graph metric where the removal of an edge incurs substantial changes to the distances of certain pairs of nodes (often in the neighborhood) – and these edges are especially susceptible to adversarial attacks. Depending on the graph topology, there are cases when the curvature upon convergence is not zero (for example, when the graph is sparse and tree-like the curvature converges to a negative value). Similarly the disturbance to the distances between nodes in the graph, upon the removal of a single edge, is disseminated in a global manner so all pairs distances suffer similar (small) damage due to the adversarial attack.

**Graph Re-sampling in GNN**    Instead of using the possibly poisoned input graph for training data, we re-sample a family of graphs from the Ricci flow metric and use this ensemble of graphs as the training data. The edges of a graph are sampled by imposing a Gaussian filter on each node using the Ricci flow metric distance. The use of a Gaussian kernel to convert the Euclidean distances between data points to a similarity measure is commonly used in settings that take a manifold viewpoint on input data.

---

**Algorithm 1** One GNN epoch based on Ricci flow graph restructuring

---

**input** Input adjacency matrix $A$, $\sigma$, $\beta$
    Pre-computed Ricci flow metric $F$ from $A$
    Pre-computed all-pairs Ricci flow distance metric $S$ of all nodes via $F$

    Pre-computed edge probability matrix: $P = \dfrac{1}{\sigma\sqrt{2\pi}} \exp\left(-\dfrac{1}{2}\left(\dfrac{S}{\beta\sigma}\right)^2\right)$

1: **for** $t = 1$ to $T$ **do**
2:    Sampling $A_R$ from $P$
3:    $A_R = A_R \vee A_R^T$
4:    $\widetilde{A}_R = A_R + I$
5:    $\hat{A}_R = \widetilde{D}_R^{-1/2}\widetilde{A}_R\widetilde{D}_R^{-1/2}$
6:    Train and update weight parameter $W^t$
7: **end for**

---

Specifically, we first calculate the edge weights $F$ by running Ricci flow on the attacked graph $G$. Then the all-pairs Ricci flow distance metric $S$ between any two nodes are calculated as the geodesic distance based on the weights assigned to all the edges. To keep graph sparsity, we only sample edges between pairs that are within $k$ hops of each other in $G$ (we take $k = 2$ in the experiments). Two nodes is connected by an edge with probability: $P(S) = \dfrac{1}{\sigma\sqrt{2\pi}} \exp\left(-\dfrac{1}{2}\left(\dfrac{S}{\beta\sigma}\right)^2\right)$. In each epoch, we sample a graph $G_i$ for each layer $i$ in the graph convolution method and apply classical GCN to learn weight parameter $W$ for prediction. The pseudo code for one GNN epoch is shown in Algorithm 1.

**Intuition.** Running Ricci flow provides us a chance to recover edges that should exist but are not formed yet (or removed by the attacker). Recall that a new graph is sampled at each layer in the graph convolution pipeline. If an edge is added from the attack and is not aligned with the main network structure (and the underlying metric space), it is unlikely to get consistent support in the re-sampling phase across multiple layers. If an edge from the attack is actually aligned well with the underlying metric space, it is not creating much damage in the performance. This observation is visualized in Figure 4. To show the benefit of Ricci flow metric in addition to the ensemble approach, we also compared with re-sampling using other graph metrics such as spectral embedding and hop count metric. See the experiment section.

## 3 EXPERIMENTS

**Attack and defense baselines.** We test our methods against two poisoning attacks: RAND (randomly adding fake edges into the graph, provided by DeepRobust (Li et al., 2020) library) and META (meta-learning attack (Zügner & Günnemann, 2019a)), which treats the graph structure as a hyper-parameter and uses meta-gradients to solve the believe optimization problem in Eq. 2.1. We ran the meta-learning with self-training (using predicted labels on unlabeled nodes) and exact meta-gradients named as *Meta-Self* since it achieves state-of-the-art performance in most datasets.

For defense methods, we also compare the performance of graph attention network (GAT) (Veličković et al., 2018), RGCN (Zhu et al., 2019), GCN-Jaccard (Wu et al., 2019), GCN-SVD (Entezari et al., 2020) and Pro-GNN (Jin et al., 2020b). The detailed descriptions of these baseline methods can be found in the supplementary materials. In all experiments, for each graph, we run the training and inference tasks 20 times and take the average accuracy. For each training procedure, we run 100 epochs and use the best model based on validation performance.

**Results on synthetic datasets.** We evaluate our method on synthetic graphs generated from the Stochastic Block Model (SBM) (Holland et al., 1983). We create 24 random graphs, each of which has 1000 nodes, equally partitioned into five communities. Within each community, two nodes are connected with intra-class probability $p \in \{0.07, 0.09, 0.11, 0.13\}$. Nodes from different classes have a lower probability to connect with each other with inter-class edge probability $q \in \{0.025, 0.03, 0.035, 0.4, 0.045, 0.05\}$. Community 4 and 5 only have edges to community 1. For

each generated graph, we randomly select 100 nodes as the training set and the remaining 900 as testing set for attack and defense. We assign each node a node ID feature using the one-hot encoding.

We use 2-layer GCN (Kipf & Welling, 2016) as the default graph neural network and meta-learning attacks (Zügner & Günnemann, 2019a) as the attacking method. 5% of the total edges are perturbed under the node degree distribution constraint as in (Zügner et al., 2018). For attacking, we follow the setting in (Zügner & Günnemann, 2019a). For Ricci-GNN, we also use 2 layers, and the hyperparameters are chosen from $\sigma \in \{0.2, 0.4, 0.6, 0.8, 1.0\}$ and $\beta \in \{1, 2, 3, 4, 5\}$.

The classification accuracy for the original graph, the attacked graph, and our Ricci-GNN method are $84\%$, $82\%$, and $87\%$ respectively. Our proposed method successfully negates the impact of the attack and even improves classification accuracy. This is due to the power of the Ricci flow metric in terms of recovering the underlying community structure and the improved robustness and diversity with our re-sampling method. Figure 4(a) shows a graph from the SBM with inter-class probability $q = 0.045$ and intra-class probability $p = 0.09$. The meta-learning attack injected additional edges (see the red dot in its adjacency matrix in Figure 4(b)), many of which are between different communities as this will incur most damage to the performance. By sampling via a Gaussian filter on the Ricci flow metric (see probability in Figure 4(c)), we can re-create graph topology that preserves the same connectivity pattern within communities and across communities. In addition, if we sample twice and obtain two graphs $G_1$ and $G_2$. Both graphs strongly preserve the community structure; but the inter-communities edges in $G_1$ and $G_2$ are largely different – see the common edges in Figure 4(d). Recall that we plug in a freshly re-sampled graph in each layer of the GCN, the influence of inter-community edges introduced by the attack is gradually eliminated during training.

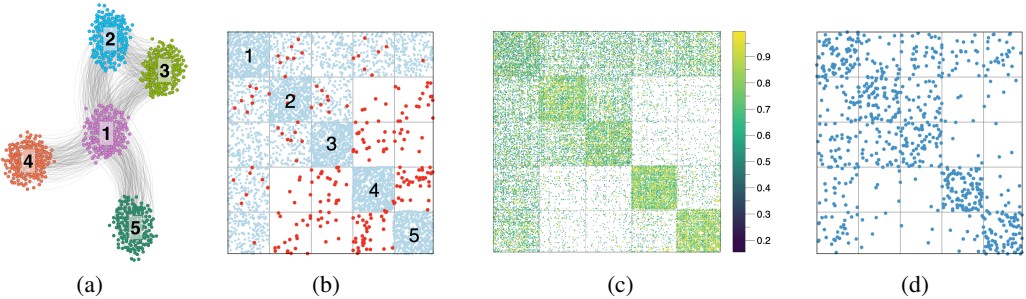

|      |      |      |      |
|------|------|------|------|
| (a)  | (b)  | (c)  | (d)  |

Figure 4: The defense of meta-learning attack using Ricci-GNN on a Stochastic Block Model (SBM) graph with 5 communities. 4(a): The clean SBM graph. 4(b): The adjacency matrix of the graph while the edges in the clean graph are shown in blue. Edges added by the meta-learning attack are shown in red. These edges appear disproportionately to the original edge density. Sparser blocks in the clean graph receive more edges in the attack; nearly all edges in the attack are between different communities. 4(c): The heat map of the probability of an edge being connected in our re-structuring method. The probability of edges in the original community structure are higher than the attack edges. 4(d): The common edges of two randomly re-sampled graphs – the influence of the edges in the meta-learning attack is essentially eliminated.

**Results on real-world datasets.** We evaluate our method on three real-world graph datasets: Cora, Citeseer, and Polblogs, that were often used in prior work (Zügner & Günnemann, 2019a). Cora and Citeseer (Sen et al., 2008) are citation networks, where each node represents a document and each edge represents a citation relationship. In Polblogs (Adamic & Glance, 2005), nodes are political blogs in 2004 US president election and edges represent citations between blog. Note that Polblogs does not have node features so we create an $N$ dimension one hot feature for each node.

We use the same training setup for the clean graph, poisoned graph and our method. This includes setting L2 regularization with $\lambda = 0.0005$, initializing by Glorot initialization and training by minimizing cross-entropy loss using Adam optimizer with learning rate $r = 0.005$.

For random attack, we randomly add $5\%$ extra edges. The result is shown in Table 1. Our method is on par with other methods on Cora and achieve state-of-the-art for Citeseer and Polblogs dataset. We also ran the experiment with increasing perturbation ratio on Polblogs dataset. As shown in Figure 5(b), by restructuring and sampling the graph based on the Ricci flow metric, our method can negate most of the effect of added random noise even when the noise ratio is large.

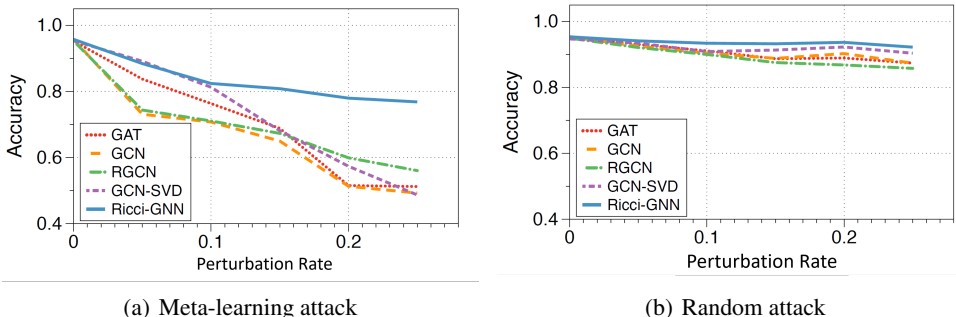

(a) Meta-learning attack          (b) Random attack

Figure 5: Accuracy plot of different defense schemes with increasing perturbation rate under different attacks on Polblogs.

Table 1: Classification accuracy for various defense schemes after random attack of 5% extra edges

| Dataset | GCN | GAT | RGCN | GCN-Jaccard | GCN-SVD | Pro-GNN | Ricci-GNN |
|---------|-----|-----|------|-------------|---------|---------|-----------|
| **Cora** | **83.54**±0.34 | 83.27±0.28 | 83.51±0.24 | 83.09±0.36 | 72.94±0.46 | 83.24 ±0.76 | 83.19±0.64 |
| **Citeseer** | 68.75±0.48 | 68.76±0.57 | 71.08 ±1.10 | 71.34±0.59 | 64.78±1.11 | 70.78±1.02 | **71.47** ±0.53 |
| **Polblogs** | 92.49 ±0.94 | 93.03 ±1.32 | 92.07 ±0.28 | - | 93.39 ±0.33 | 93.28±0.91 | **94.10** ±0.26 |

Table 2: Classification accuracy for various defense schemes after meta-learning attack

| Dataset | Ptb rate% | GCN | GAT | RGCN | GCN-Jaccard | GCN-SVD | Pro-GNN | Ricci-GNN |
|---------|-----------|-----|-----|------|-------------|---------|---------|-----------|
| **Cora** | 0 | 83.50±0.44 | **83.97**±0.65 | 83.09±0.44 | 82.05±0.51 | 80.63±0.45 | 82.98±0.23 | 83.03 ±0.59 |
| | 5 | 76.55±0.79 | 80.44±0.74 | 77.42±0.39 | 79.13±0.59 | 78.39±0.54 | 82.78 ±0.45 | **82.80** ±0.43 |
| | 10 | 70.39±1.28 | 75.61±0.59 | 72.22±0.38 | 76.16±0.76 | 71.47±0.83 | 77.91±0.86 | **79.70** ±0.43 |
| | 15 | 65.10±0.71 | 69.78±1.28 | 66.82±0.39 | 71.03±0.64 | 66.69±1.18 | 76.01 ±1.12 | **77.28**±0.50 |
| | 20 | 59.56±2.72 | 59.94±0.92 | 59.27±0.37 | 65.71±0.89 | 58.94±1.13 | 68.78 ±5.84 | **74.10** ±0.63 |
| | 25 | 47.53±1.96 | 54.78±0.74 | 50.51±0.78 | 60.82±1.08 | 52.06±1.19 | 56.54 ±2.58 | **71.73**±0.60 |
| **Citeseer** | 0 | 71.96±0.55 | 73.26±0.83 | 71.20±0.83 | 72.10±0.63 | 70.65±0.32 | 73.26 ±0.69 | **73.95**±0.53 |
| | 5 | 70.88±0.62 | 72.89±0.83 | 70.50±0.43 | 70.51±0.97 | 68.84±0.72 | 73.09 ±0.34 | **73.39**±0.52 |
| | 10 | 67.55±0.89 | 70.63±0.48 | 67.71±0.30 | 69.64±0.56 | 68.87±0.62 | 72.43 ±0.75 | **72.51**±0.62 |
| | 15 | 64.52±0.62 | 69.02±0.62 | 65.69±0.62 | 65.95±0.62 | 63.26±0.62 | 70.82 ±2.38 | **71.99**±0.71 |
| | 20 | 62.03±3.49 | 61.04±1.52 | 62.49±1.22 | 59.30±1.40 | 58.55±1.09 | 66.19 ±2.57 | **68.40**±0.51 |
| | 25 | 56.94±2.09 | 61.85±1.12 | 55.35±0.66 | 59.89±1.47 | 57.18±1.87 | 66.40 ±2.57 | **68.84**±0.51 |
| **Polblogs** | 0 | 95.69±0.38 | 95.35±0.20 | 95.22±0.14 | - | 95.31±0.18 | 93.20 ±0.64 | **95.72**±0.24 |
| | 5 | 73.07±0.80 | 83.69±1.45 | 74.34±0.19 | - | 89.09±0.22 | **93.29** ±0.18 | 90.54 ±0.47 |
| | 10 | 70.72±0.62 | 76.32±0.62 | 71.04±0.62 | - | 81.24±0.62 | 89.42 ±1.09 | 86.88±0.85 |
| | 15 | 64.96±1.91 | 68.80±1.14 | 67.28±0.38 | - | 68.10±3.73 | 86.04 ±2.21 | **86.10**±0.96 |
| | 20 | 51.27±1.23 | 51.50±1.63 | 59.89±0.34 | - | 57.33±3.15 | 79.56 ±5.68 | **81.37**±1.24 |
| | 25 | 49.23±1.36 | 51.19±1.49 | 56.02±0.56 | - | 48.66±9.93 | 63.18 ±4.40 | **79.95** ±1.89 |

Table 2 shows the classification accuracy (the higher the better) of different defense schemes after attacked by meta-learning attack with different perturbation ratios. The first row of the table in each dataset shows the accuracy of applying the defense method on the original clean graph. The result shows that directly using our method has no negative effect on GNN models. Specially for Citeseer and Polblog dataset, our method achieve state-of-the-art accuracy. When the perturbation increases (the attacker is more powerful), the accuracy gap between our method and other methods widens, clearly demonstrating the advantage of our method.

Table 3: Classification accuracy for defense using Ricci flow metric (Ricci) vs. hop count metric (HC) and spectral embedding metric after meta-learning attack. Note that Citeseer dataset is by itself fairly robust to attacks (See Table 2).

| Ptb rate % | Cora | | | Citeseer | | | Polblogs | | |
|:---:|:---:|:---:|:---:|:---:|:---:|:---:|:---:|:---:|:---:|
| | HC | Spectral | Ricci | HC | Spectral | Ricci | HC | Spectral | Ricci |
| **0** | 82.0 | 78.4 | **83.0** | 73.4 | 72.1 | **74.0** | 94.9 | 93.4 | **95.7** |
| **5** | 81.4 | 76.1 | **82.8** | 73.0 | 71.3 | **73.4** | 85.6 | 88.2 | **90.5** |
| **10** | 77.5 | 73.1 | **79.7** | 71.0 | 69.3 | **72.5** | 85.6 | **88.0** | 86.9 |
| **15** | 75.5 | 65.1 | **77.3** | 70.7 | 67.6 | **72.0** | 70.6 | 80.3 | **86.1** |
| **20** | 72.8 | 60.2 | **74.1** | 67.9 | 67.8 | **68.4** | 65.8 | 79.4 | **81.3** |
| **25** | 65.4 | 54.7 | **71.7** | 65.4 | 65.6 | **68.8** | 62.0 | 74.6 | **79.5** |

**Benefit of Ricci flow metric.** To show the importance of Ricci flow metric, we also run the entire graph restructuring algorithm but with the hop count metric and the spectral embedding metric against meta-learning attacks. See Table 3. In nearly all cases, using Ricci flow metric shows clear improvement on performance. On Cora, the one with spectral embedding is the worst, as the graph is relatively sparse and spectral embedding is least stable. On Polblogs, the one with hop count distance is the worst, as the graph is very dense and diameter is small (only 4). It shows that the Ricci flow metric is important for the probabilistic sampling framework to achieve the full defense potential.

## 4 CONCLUSION

We propose a novel approach to improve the robustness of GNNs against graph-topology focused attacks. The curvature and flow information can effectively capture the intrinsic geometry of the graph that is robust to structural perturbation. Our algorithm restructures and resamples the graphs using the underlying geometry. This helps training a robust graph neural network. Our method achieve superior performance on both synthetic and real work benchmarks under various attacks.

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

## A   APPENDIX

In this supplemental material, we provide technical details on Ricci curvature and flow. We also provide additional details on the experiments.

## A   TECHNICAL DETAILS OF RICCI CURVATURE AND FLOW

In this sub-section, we will establish the precise mathematical formulation of Ricci curvature and Ricci flow in the discrete setting and describe their computation on un-directed graphs. We first define Ricci curvature, which is computed for each edge. Next, we explain how Ricci flow re-weight the edges iteratively so that Ricci curvatures of all edges are smoothed. The final weighted graph induces the Ricci flow metric; the geodesic distance between any two nodes in this weighted graph is their distance in the Ricci flow metric space.

### A.1   DISCRETE RICCI CURVATURE.

In Ollivier's definition Ollivier (2009) of Ricci curvature for discrete space, one aims to measure the curvature $\kappa_{xy}$ between nodes $x$ and $y$. By comparing the Wasserstein distance (also called earth mover distance) between the neighborhoods of $x$ and $y$, we can determine the deviation of the edge $(x, y)$ from being flat.

For an undirected and edge-weighted graph $G = (V, E)$, the neighborhood of a node $x$ is the collection of immediately adjacent nodes (one-hop neighbors) $\mathcal{N}(x) = \{x_i : (x, x_i) \in E\}$ associated with some probability measure $m_x(x_i)$ which sums to 1. Similarly, we have a probability measure $m_y$ on the neighborhood of $y$. The Wasserstein distance between thew two probability measures, $W(m_x, m_y)$, is the minimum total weighted cost to move $m_x$ to $m_y$ using the optimal transportation plan $\mathbb{M}$:

$$
\begin{aligned}
\min_{\mathbb{M}} \quad & \sum_{i,j} d(x_i, y_j) \mathbb{M}(x_i, y_j) \\
\text{s.t.} \quad & \sum_{j} \mathbb{M}(x_i, y_j) = m_x(x_i), \forall i \\
& \sum_{i} \mathbb{M}(x_i, y_j) = m_y(y_j), \forall j
\end{aligned}
\tag{A.1}
$$

where $\mathbb{M}(x_i, y_j)$ is the quantity of probability mass transferred from $x_i$ to $y_j$ using the shortest path with graph geodesic $d(x_i, y_j)$. Then the Ricci curvature $\kappa_{xy}$ of the edge $(x, y)$ takes the ratio between this Wasserstein distance and their geodesic:

$$
\kappa_{xy} = 1 - \frac{W(m_x, m_y)}{d(x, y)}.
\tag{A.2}
$$

A negative curvature means the probability mass of the neighborhood $m_x$ is transported to $m_y$ mostly through the edge $(x, y)$. This usually happens when $(x, y)$ is a bridge joining two communities (red edges in Figure 1(a)). Meanwhile, an edge within a community tend to have overlapping neighborhoods of $x$ and $y$, resulting in positive curvature (blue edges in Figure 1(b)). Notice that the curvature value depends on the edge weights. Later in Ricci flow, we will illustrate how curvature morphs when edge weights change.

To define the probability measure of each neighborhood, we adopt a weight aware definition from Ni et al. (2019) that discounts neighbors which are further away. With portion $\gamma \in [0, 1]$ and discount factor $p \geq 0$,

$$
m_x^{\gamma, p}(x_i) = \begin{cases} \gamma & \text{if } x_i = x \\ \frac{1-\gamma}{C} \cdot \exp(-d(x, x_i)^p) & \text{if } x_i \in \mathcal{N}(x) \\ 0 & \text{otherwise .} \end{cases}
$$

where $C = \sum_{i:x_i \in \mathcal{N}(x)} \exp(-d(x, x_i)^p)$ is the normalizing constant to assign probability measure 1 to the entire neighborhood. When $p = 0$ (weight unaware), this definition of the probability measure reduces to the uniform distribution. We take $\gamma = 0.5$ and $p = 2$ following the heuristics in the literature.

## A.2  RICCI FLOW

Recall the curvature describes the degree of a surface being curved. Ricci flow is an iterative process to restore the flatness everywhere such that Ricci curvature $\kappa_{xy}$ is constant for every edge. Starting with the original input graph, the flow iteratively updates edge weights. For the $t$-th iteration, all the new edge weights $w^{(t+1)}$ are calculated as

$$w^{(t+1)}(x, y) = d^{(t)}(x, y) - \kappa_{xy}^{(t)} d^{(t)}(x, y). \tag{A.3}$$

.

Each update moves the edge weight in the opposite direction of the curvature. Geometrically, negatively curved edges acting as bridges will be extended while positively curved edges within the community will be shortened. We re-normalize edge weights after each iteration. When the process converges, the final set of edge weights induces a Ricci flow metric on the graph. Please see Figure 1(b) for an illustration.

To speed up the computation of the Wasserstein distance, we use an approximate version called Sinkhorn distance from Cuturi (2013) that smooths the optimal transportation cost with a regularization term and then can be computed by Sinkhorn-Knopp's matrix scaling algorithm.

# B  ADDITIONAL EXPERIMENTAL DETAILS AND RESULTS

## B.1  DATASETS DESCRIPTION

Table 4 provides details of the datasets.

Table 4: Statistics of Real-World graph datasets.

|  | **Cora** | **CiteSeer** | **Polblogs** |
|---|---|---|---|
| **#Node** | 2708 | 3327 | 1222 |
| **#Edge** | 5429 | 4732 | 16714 |
| **AvgDeg** | 4.00 | 2.84 | 27.35 |
| **Diameter** | 19 | 28 | 4 |
| **#Feature** | 1433 | 3703 | - |
| **#Class** | 7 | 6 | 2 |

## B.2  GRAPH HEATMAP

Figure 6 shows prediction accuracy of all the 24 graphs generated in Section 4.1 of the main paper. We observe better accuracy by our method, compared with the GNN trained on the attacked graph.

## B.3  ADDED BASELINES

### B.3.1  NEW BASELINES RESULT ON DEFENDING META-LEARNING ATTACK FOR DIFFERENT PERTURBATION RATE.

Tabel 5 shows the two new baselines performance on defending the meta learning attack under different perturbation rate. It worth mentioning that CurvGN (Ye et al., 2020) is not designed for improving the robustness of GNN. Thus CurvGN performs relatively worse than DropEdge (Rong et al., 2019). We also move the performance of our Ricci-GNN to here to show that our method perform better than those two methods.

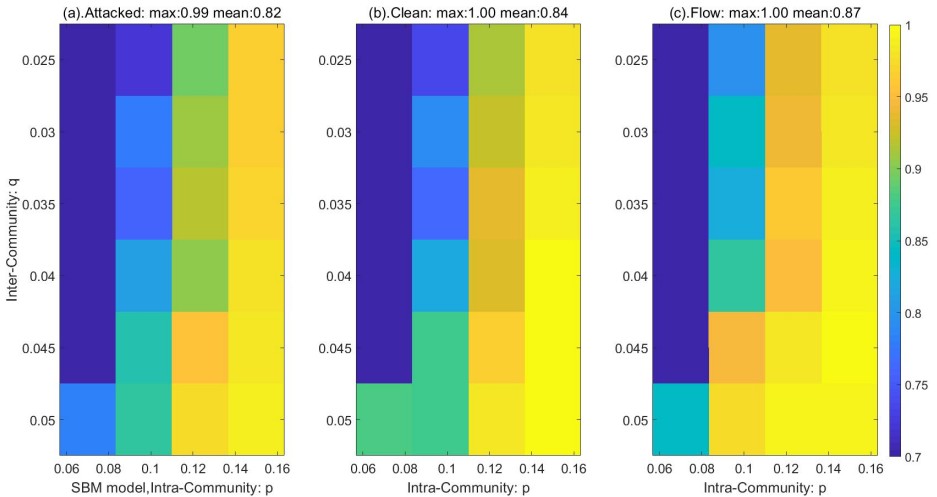

Figure 6: Defense accuracy heat maps for synthetic data of 24 SBM graphs constructed from different $\{p, q\}$. From left to right: attacked graph, clean graph, defense result by our Ricci-GNN. For each heat map, x-axis is the intra-community probability, $p$, y-axis is the inter-community probability, $q$.

Table 5: Classification accuracy for various defense schemes after meta-learning attack

| Dataset | Ptb rate% | CurvGN | DropEdge | Ricci-GNN |
|---------|-----------|--------|----------|-----------|
| **Cora** | 0 | 82.8 | 82.8 | **83.0** |
| | 5 | 75.8 | 79.2 | **82.8** |
| | 10 | 69.3 | 72.3 | **79.7** |
| | 15 | 60.3 | 65.9 | **77.3** |
| | 20 | 52.8 | 57.9 | **74.1** |
| | 25 | 46.7 | 47.8 | **71.7** |
| **Citeseer** | 0 | 73.4 | 73.4 | **73.9** |
| | 5 | 70.5 | 71.9 | **73.4** |
| | 10 | 66.1 | 69.3 | **72.5** |
| | 15 | 62.9 | 65.9 | **72.0** |
| | 20 | 52.9 | 58.1 | **68.4** |
| | 25 | 48.1 | 55.6 | **68.8** |
| **Polblogs** | 0 | 94.1 | 94.4 | **95.7** |
| | 5 | 73.7 | **90.7** | 90.5 |
| | 10 | 72.2 | **89.0** | 86.9 |
| | 15 | 68.2 | 82.2 | **86.1** |
| | 20 | 58.5 | 81.3 | **81.4** |
| | 25 | 55.6 | 78.8 | **80.0** |

### B.3.2 DEFENSE RESULT ON TOPOLOGICAL ATTACK-MINMAX

Besides the Table 6 shows the extra experiment results on defending the topological attack-MinMax (Xu et al., 2019a) under two different perturbation rate: 5% and 25%. Since the code of DropEdge has programming error on citeseer under MinMax attack, we can't report it's result. (For now) From the tabel, we can see that our methods out perform all other baselines. It confirms that our method improved the robustness of GNN on different attack methods.

Table 6: Classification accuracy for various defense schemes after topological attack-MinMax.

| Dataset | Ptb rate% | GCN | GAT | RGCN | GCN-Jaccard | GCN-SVD | Pro-GNN | DropEdge | curvGN | Ricci-GNN |
|---------|-----------|------|------|------|-------------|---------|---------|----------|--------|-----------|
| Cora | 5 | 82.8 | 83.8 | 81.9 | 71.3 | 83.5 | 83.4 | 80.3 | 82.3 | **83.8** |
|  | 25 | 74.2 | 73.5 | 77.3 | 65.5 | 75.9 | 77.2 | 44.5 | 72.6 | **78.6** |
| Citeseer | 5 | 71.6 | 72.2 | 73.3 | 67.8 | 71.6 | 71.7 | - | 72.3 | **73.7** |
|  | 25 | 58.7 | 59.3 | 63.5 | 56.0 | 57.5 | 66.4 | - | 67.7 | **68.4** |
| Polblogs | 5 | 88.5 | 87.1 | 51.2 | 86.7 | 71.2 | 92.7 | 80.4 | 93.6 | **94.5** |
|  | 25 | 58.5 | 53.0 | 51.4 | 50.2 | 72.2 | 64.9 | 59.4 | 75.6 | **87.9** |

## C   DETAILED DESCRIPTION OF BASELINE METHODS

**GAT**. Graph attention network (Veličković et al., 2018) uses node feature to learn a self attention. The attention is used to re-weight each message passed to the node. Neighboring nodes with features that are more important will receive higher weights. Since the message is solely learned from node features, GAT is inherently robust to graph structure perturbation.

**RGCN**. RGCN (Zhu et al., 2019) models hidden layer representation of nodes as Gaussian distribution to counter adversarial attacks. It also use attention mechanism from GAT to penalize nodes with high variance.

**GCN-Jaccard**. GCN-Jaccard (Wu et al., 2019) inherits the idea of feature importance from GAT. It choose important messages measured by Jaccard similarity of features and delete edges that considered irrelevant.

**GCN-SVD**. GCN-SVD (Entezari et al., 2020) claims that most adversarial attacks will affect high-rank spectrum of the graph, thus taking a low-rank approximation of the graph to defend the adversarial attacks. Note that it's originally designed to defend Nettack(Zügner et al., 2018). However, it can be also used for meta-learning attack (Zügner & Günnemann, 2019a) and random attack.

**Pro-GNN**. Pro-GNN jointly learn a structural graph and a robust graph neural network model from the perturbed graph guided by exploring the graph properties of sparsity, low rank and feature smoothness to design robust graph neural networks.

**Topological attack-MinMax**. Topological attack provides two attacking methods. a) attacking a pre-defined GNN (PGD) and b) attacking a re-trainable GNN (MinMax). We choose the MinMax because most recent work focus on improve the robustness of re-trained GNN. The topological attack the problem as a loss optimization problem and use the MinMax method to solve the problem. The attacker seeks to minimize the per-node attack loss while the GNN tries defend the attack by retraining W so that attacking GNN is more difficult.

