# OpenReview forum: "Ricci-GNN: Defending Against Structural Attacks Through a Geometric Approach"
_ICLR.cc/2021/Conference — Reject_

### Official Review · AnonReviewer2 · 2020-10-27
**The robustness evaluation could be improved**

**Rating:** 6
**Confidence:** 5

**Review:**

Summary:
In Ricci-GCN new graphs are resampled in each iteration of the training phase based on the Ricci flow metric. The Ricci flow incorporates curvature information and captures the intrinsic geometry of the graph. Compared to e.g. spectral embedding it is more robust to structural perturbations. This leads to improved robustness against adversarial attacks on the graph structure.

Reasons for score:
Overall, I vote for accepting. The idea is well motivated, the paper is well written, and the experiments show a clear increase in robustness on real data. My major concern is not using an adaptive attack to evaluate robustness (see weak points).

Strong points:
* The main idea of using Ricci flow is interesting, well motivated and well executed.
* Evaluation on SBM graphs helps with better understanding why the proposed approach works.
* The comparison to other metrics (Spectral and HC) is appreciated.

Weak points:
* It is not clear whether META (the meta-learning attack) is computed w.r.t. the vanilla GCN or the Ricci-GCN. If META was run on the original GCN it is not clear whether the attack is not successful because Ricci-GCN is more robust or because the adversarial edges found for GCN are not transferable to Ricci-GCN. Since the proposed defense is only heuristic (not certifiable) in order to show robustness it has to be evaluated against an adaptive attacker that takes the defense into account [1]. Otherwise, an adaptive attacker may easily break the defense in the future. For example, META can be adapted to account for the Ricci flow. If META was indeed run on Ricci-GCN, the author should discuss the details, e.g. whether they use the reparametrization trick to compute the gradients through the sampling.
* The gain in robustness is only significant for large perturbation rates (>0.1) which might not correspond to realistic threat models in practice, e.g. for perturbation rates <0.1 GCN-SVD is on par with Ricci-GCN.
* One interpretation of the proposed approach is that Ricci-GCN is doing (a specific type of) data augmentation which is known to improve generalization and by extension the clean and the adversarial accuracy. For example, one augmentation in [2] is to randomly sample edges to add or remove, and in [3] edges are randomly dropped. Even though [2] and [3] are not motivated by robustness they are a relevant baselines since similar to Ricci-GCN they generated different graphs during training.
* A big drawback of the proposed approach is the large number of hyperparameters: \gamma=0.5, p=2, k=2, \sigma, \beta, etc. It this is not clear how sensitive is the method to these choices or to the definition of the "probability measure of each neighborhood".
* Ignoring the random attack which extremely weak, the evaluation is limited to a single attack (META).  Evaluating against other attacks (see [4] and [5]) would help to better evaluate the robustness of the model.

Question for the authors:
1. How robust is Ricci-GCN to adaptive attacks? (see weak points)
2. How does Ricci-GCN compare to other data augmentation techniques (see weak points)
3. Is sampling performed only during training? If so are there any benefits to also sampling during inference (and aggregating the predictions)? Can sampling during inference help defend against evasion attacks such as Nettack?
4. Is there any improvement if we are willing to pay the price of decreased sparsity, take k larger than 2?
5. Why is the row for perturbation rate of 0 omitted from Table 3? Where the hyperparameters tuned separately for Spectral and HC?

Additional feedback that did not affect the decision:
* The related work should also discuss the difference between certifiable and heuristic defenses and how the proposed approach fits in this context.
* It would be nice to quantitatively show "The probability of edges in the original community
structure are higher than the attack edges"
* It would be beneficial to provide a reference or evidence for the claim "generally lacks descriptive power to provide desirable resolution and differentiation". While the results in Table 3 provide indirect evidence, it is not clear that this is due to "lack of descriptive power".
* It would be interesting to see whether Ricci-GCN is also more certifiably robust than vanilla GCN, e.g. by computing model agnostic certificates such as [6].

Typos:
* Figure 5 Captions: Purterbation Rate

## After Rebuttal
The authors' response clarified some of the issues and partially addressed some of my concerns. Based on this and the remaining reviews I have decided to keep the score unchanged.

One additional comment regarding the evaluation: In the authors' response they state "Finally, we would like to point out that it is common practice to use GCN as a subroutine for Meta-attack against different defense methods. This was shown in the original Meta-Attack paper, as well as multiple follow-up defense papers." I would like to again point out that the fact that this is a common practice is not ideal, even though multiple follow-up defense papers use the same strategy. We have already learned the lesson in the computer vision literature that adaptive attacks are the least we can do to evaluate heuristic defenses (see [1]) and even that might not provide strong evidence.

References:
1. Tramer, Florian, Nicholas Carlini, Wieland Brendel, and Aleksander Madry. "On adaptive attacks to adversarial example defenses."
2. Wang, Yiwei, Wei Wang, Yuxuan Liang, Yujun Cai, Juncheng Liu, and Bryan Hooi. "NodeAug: Semi-Supervised Node Classification with Data Augmentation."
3. Rong, Yu, Wenbing Huang, Tingyang Xu, and Junzhou Huang. "Dropedge: Towards deep graph convolutional networks on node classification."
4. https://github.com/gitgiter/Graph-Adversarial-Learning
5. https://github.com/safe-graph/graph-adversarial-learning-literature
6. Bojchevski, Aleksandar, Johannes Klicpera, and Stephan Günnemann. "Efficient robustness certificates for discrete data: Sparsity-aware randomized smoothing for graphs, images and more."

---

> ### Author Response · Authors · 2020-11-20
> **Add clarification of some issues and add baselines as the reviewer suggested**
>
> Weak points:
>
> **[1](It is not clear...)**
>
> **Ans**: META learning uses GCN as a subroutine. We apply Ricci-GCN on the graph containing the contaminated/adversarial edges.
>
> **[2](The gain in…)**
>
> **Ans**: You are absolutely right. Since the attacker’s power is strictly limited when the perturbation rate is low. The difference between different defense methods and even the base method (GCN and GAT) is not large, but our method still outperforms the other baselines on almost all perturbation rates.
>
> **[3](One interpretation…)**
>
> **Ans**: Thank you for pointing it out. We add the DropEdge as another baseline and show the new results in Appendix (B.3.2).
>
> **[4](A big drawback…)**
>
> **Ans**: \sigma and \beta: for each parameter, we tested about five values. In our experiments, we did not run parameter search on parameters \gamma, p, k. Below is the rationale for the choice of these parameters.
> The choice of k=2: we resample edges among the 2-hop neighborhood in the original graph. We fix k=2 for two reasons: 1) due to the small world property of most random graphs and real-world graphs the diameter is a small constant; 2) two nodes of three or more hops away are expected to have a long geodesic distance, and thus the chance of being sampled is low. \gamma and p are parameters used in the computation of Ricci flow. In prior work [Ni 2019], the influence of these parameters on the computed curvature has been thoroughly evaluated. In short, the choice of different \gamma mainly introduces a global shift and scaling of the curvature values. The choice of p influences the convergence rate. In our experiments, we fixed the value of \gamma and p as the suggested values as in [Ni 2019]. We also include a few new figures with different values of \gamma and p, which shows that there is no significant difference in a qualitative manner. Similarly, the choice of probability measure used in the curvature definition has been fully investigated in the prior work [Ni 2018] and [Ni 2019], which suggested to use the exponential distribution. This choice is also fixed in the experiments.
>
> **[5](Ignoring the random...)**
>
> **Ans**: We add another SOTA attack algorithm Topological Attack-MinMax (from paper “Topology Attack and Defense for Graph Neural Networks: An Optimization Perspective”). The results are shown in the Appendix (Table 5). Our Ricci-GNN achieves the best performance on all datasets under this new attack.
>
> Question for the authors:
>
> **[1](How robust is…)**
>
> **Ans**: Ricci curvature, defined on an edge, can be considered as a measure of how much the removal of one edge changes the network connectivity. Edges of negative curvature (e.g., a critical edge connecting two communities), if deleted, can possibly completely change the global network layout. Ricci flow (which is used in this paper) is to modify the edge weights such that all edges have the same curvature. The new weights are observed in [Ni 2018] to be more robust to *random* edge insertion/deletion, compared to alternative network metrics. This inspired us to apply Ricci flow metric to improve robustness to *adversarial attacks* in graph learning.
>
> **[2](How does…)**
>
> **Ans**: Ricci-GNN is more robust than other data augmentation techniques in most cases.
>
> **[3](Is sampling…)**
>
> **Ans**: No, it’s sampled every time an adjacency matrix is needed in both training and testing stages.
>
> **[4](Is there any…)**
>
> **Ans**: If we increase k too much, there will be an over smoothing problem. And the computation and memory cost will be significantly increased.
>
> **[5](Why is the row...)**
>
> **Ans**: Thank you for pointing it out. Originally, we think that the main purpose is to compare the robustness of different approaches. We will add the 0 perturbation rate to the paper. Ricci flow metric is better than both hop count metric (HC) and metric from spectral embedding (Spectral).

---

> > ### Comment · AnonReviewer2 · 2020-11-23
> > **Clarification**
> >
> > Thank you for your reply. Your answers clarified some of my questions. I have two follow up questions:
> >
> > I am not sure if the first bullet point in the weak points (and the related first question) was clear enough, so I will try to rephrase it. As you said the META attack uses GCN as a subroutine. If you apply Ricci-GCN on the graph containing the adversarial edges found from this default version of META there are two explanations for the results:
> > * The adversarial edges found using GCN as a subroutine are not *transferable* to Ricci-GCN, i.e. the attack is weak, in which case we cannot tell if Ricci-GCN is really more robust
> > * They are transferable and Ricci-GCN is indeed more robust
> >
> > Unfortunately, we cannot tell these two explanations apart based on data from the current experiments. Alternatively, one can easily extend META to use Ricci-GCN as a subroutine (making sure the final loss in META is still differentiable despite the sampling). This will make the attack adaptive and stronger.
> >
> > 1. How well does Ricci-GCN perform against an adaptive attack as described above?
> >
> > The second question is related to evasion attacks:
> >
> > 2. How well does Ricci-GCN perform against evasion attacks such as Nettack?

---

> > > ### Author Response · Authors · 2020-11-24
> > > **Answering the Two Questions Regarding Attacks**
> > >
> > > Thank you for the response. We address your two questions below.
> > >
> > > **Q1:** Transferability of GCN-based Meta-attack. How well does Ricci-GCN perform against an adaptive attack as described above?
> > >
> > > **Ans:** Good point. However, adapting Meta-Attack to directly attack Ricci-GNN is not easy. Ricci-GNN has two critical components: graph sampling for training, and an iterative PDE-like process to calculate Ricci flow metric which flattens the curvatures everywhere. Both parts are technically very challenging for a gradient-based attack (e.g., Meta-Attack) to propagate through. One might have to resort to various advanced techniques such as reinforcement learning and Gumbel-softmax. This requires significant new contributions which are beyond the scope of this paper.
> > >
> > > Even if we assume the technical challenges have been resolved, we believe Ricci-GNN is indeed robust against even specifically designed attacks. The intrinsic robustness of Ricci flow ensures that under a given budget of edge insertion/deletion, the Ricci flow metric and thus the probability matrix derived from this metric do not change significantly. (For intuition of the robustness, see our response to AnonReviewer1 and “robustness of Ricci flow metric” in Section 2.2.) Therefore, any graph-topology-based attack will be ineffective in changing the metric/probability matrix.
> > >
> > > The risk of Ricci-GNN is further mitigated by the random sampling strategy. Note that the attacker can only give a deterministic set of edge perturbation. Fake edges introduced by attackers are unlikely to appear consistently in graph samples (illustrated in Figure 4).
> > >
> > > Finally, we would like to point out that it is common practice to use GCN as a subroutine for Meta-attack against different defense methods. This was shown in the original Meta-Attack paper, as well as multiple follow-up defense papers.
> > >
> > > **Q2:** How well does Ricci-GCN perform against evasion attacks such as Nettack?
> > >
> > > **Ans:** As we explained in the introduction, in this paper we are focusing on non-targeted attacks. For local evasion attacks we are confident that our method can also be effective. But depending on the attacking method there may be a more theoretically certified way to do it, which is out of the scope of this paper. Since our method is reconstructing the global graph topology instead of a local neighborhood, it is easier to show its efficacy in defending against global non-targeted attacks. We demonstrated its clear advantage over other methods against two advanced attacks in our paper.

---

### Official Review · AnonReviewer1 · 2020-10-28
**The work presents a method to defend against the addition or removal of adversarial edges. The problem formulation is clear and the paper is easy to read; however, the current version of the paper has many weaknesses, so it requires revision.**

**Rating:** 5
**Confidence:** 4

**Review:**

Strengths:

The paper is well written and clean.

Weaknesses:

I have several concerns regarding this paper.

•	Novelty. The authors propose to use Ricci flow to compute the distance between nodes so that to sample edges with respect to that distance. Using Ricci flow for distance computation is a well-studied area (as indicated in related work). The only novel part is that each layer gets a new graph; however, this choice is not motivated (why not to train all layers of GNN on different graphs instead?) and has problems (see next).

•	Approach. Computing optimal transport distance is generally an expensive procedure. While authors indicated that it takes seconds to compute it on 36 cores machine, it’s not clear how scalable this method is. I would like to see whether it scales on normal machines with a couple of cores. Moreover, how do you compute exactly optimal transport, because the Sinkhorn method gives you a doubly stochastic matrix (how do you go from it to optimal transport?).

•	Algorithm. This is the most obscure part of the paper. First, it’s not indicated how many layers do you use in experiments. This is a major part of your algorithm because you claim that if an edge appears in several layers it means that it’s not adversarial (or that it does not harm your algorithm). In most of the baselines, there are at most 2-3 layers. There are theoretical limitations why GNN with many layers may not work in practice (see, the literature on “GNN oversmoothing”). Considering that you didn’t provide the code (can you provide an anonymized version of the code?) and that your baselines (GCN, GAT, etc.) have similar (or the same) performance as in the original papers (where the number of layers is 2-3), I deduce that your model Ricci-GNN also has this number of layers. With that said, I doubt that it’s possible to make any conclusive results about whether an edge is adversarial or not with 2-3 graphs. Moreover, I would expect to see an experiment on how your approach varies depending on the number of layers. This is a crucial part of your algorithm and not seeing discussion of it in the paper, raises concerns about the validity of experiments.

•	Design choices. Another potential problem of your algorithm is that the sampled graphs can become dense. There are hyperparameters \sigma and \beta that control the probabilities and also you limit the sampling only for 2-hop neighborhoods (“To keep graph sparsity, we only sample edges between pairs that are within k hops of each other in G (we always take k = 2 in the experiments).” This is arbitrary and the effect of it on the performance is not clear. How did you select parameters \sigma and \beta? Why k=2? How do you ensure that the sampled graphs are similar to the original one? Does it matter that sampled graphs should have similar statistics to the original graph? I guess, this crucially affects the performance of your algorithm, so I would like to see more experiments on this.

•	Datasets. Since this paper is mostly experimental, I would like to see a comparison of this model on more datasets (5-7 in total). Verifying on realistic but small datasets such as Cora and Citeseer limits our intuition about performance. For example, Cora is a single graph of 2.7K nodes. As indicated in [1], “Although small datasets are useful as sanity checks for new ideas, they can become a liability in the long run as new GNN models will be designed to overfit the small test sets instead of searching for more generalizable architectures.” There are many sources of real graphs, you can consider OGB [2] or [3].

•	Weak baselines. Another major concern of the validity of the experiments is the choice of the baselines. Neither of GNN baselines (GCN, GAT, etc.) was designed for the defense of adversarial attacks, so choosing them for comparison is not fair. A comparison with previous works (indicated in “Adversarial attack on graphs.” in related work section) is necessary. Moreover, an experiment where you randomly sample edges (instead of using Ricci distance) is desirable to compare the performance against random sampling.

•	Ablation. Since you use GCN, why the performance of Ricci-GCN is so different from GCN when there 0 perturbations? For Citeseer the absolute difference is 2% which is quite high for the same models. Also, an experiment with different choices of GNN is desirable.

•	Training. Since experiments play important role in this paper, it’s important to give a fair setup for the models in comparison. You write “For each training procedure, we run 100 epochs and use the model trained at 100-th epoch.”. This can disadvantageous for many models. A better way would be to run each model setup until convergence on the training set, selecting the epoch using the validation set. Otherwise, your baselines could suffer from either underfitting or overfitting.

[1] https://arxiv.org/pdf/2003.00982.pdf
[2] https://ogb.stanford.edu/
[3] https://paperswithcode.com/task/node-classification

==========

After reading the authors comments.

I applaud the authors for greatly improving their paper via the revision. Now the number of layers is specified and the explanation of having many sampled graphs during training is added, which was missing in the original text and was preventing a full understanding of the reasons why the proposed approach works. Overall, I am leaning toward increasing the score.

I still have several concerns about the practicality of Ricci-GNN. In simple words, the proposed approach uses some metric S (Ricci flow) that dictates how to sample graphs for training. The motivation for using Ricci flow is “that Ricci flow is a global process that tries to uncover the underlying metric space supported by the graph topology and thus embraces redundancy”. This claim cites previous papers, which in turn do not discuss what exactly is meant by “a global process that tries to uncover the underlying metric space”. Spectral embeddings also can be considered as a global metric, so some analysis on what properties of Ricci flow makes it more robust to attacks would be appreciated. Also including random sampling in comparison would confirm that the effect is coming not from the fact that you use more graphs during the training, but from how you sample those graphs. In addition, as the paper is empirical and relies on the properties of Ricci flow which was discussed in previous works and was not addressed in the context of adversarial attacks, having more datasets (especially larger ones) in the experiments would improve the paper.

---

> ### Author Response · Authors · 2020-11-20
> **We would like to point out that there seem to be quite a few misunderstandings, including Sinkhorn, graph sampling, defense baselines, etc.(1/2)**
>
> Thank you for your comment. First, we would like to point out that there seem to be quite a few misunderstandings, including Sinkhorn, graph sampling, defense baselines, etc. Below we respond to each comment in detail:
>
> Weaknesses:
>
> **[1] (Novelty)**
>
> **Ans**: “Using Ricci flow for distance computation is a well-studied area (as indicated in related work). The only novel part is that each layer gets a new graph.”
> Ans: We do not think this paper is incremental just because we are using the existing Ricci flow metric. We are the first to apply the Ricci flow metric to GNN. Our major novelty is to customize this principled tool to address the robustness of GNN to adversarial attacks problems in an effective manner. The idea is built on deep insights into the connection between the Ricci flow theory and the GNN robustness and has demonstrated great performance improvement over prior work.
>
> **[2] (Approach)**
>
> **Ans**: “because the Sinkhorn method gives you a doubly stochastic matrix (how do you go from it to optimal transport?”
> Ans: It seems the reviewer has a misunderstanding of the “Sinkhorn” method. The Sinkhorn distance is a standard method to approximate the optimal transport distance by adding entropy regularization. In the paper, we have provided the reference of the Sinkhorn method (Cuturi (2013)) for optimal transport.  In our case, we did not compute the exact optimal transport but used the “Sinkhorn distance” based on the “Sinkhorn method” as an approximation instead to compute Ricci flow for the graph. This approximation to the exact optimal transport distance in the Ricci flow computation is shown to give similar results in the referenced paper “Community Detection on Networks with Ricci Flow.”
> The running time to compute Ricci flow (https://github.com/saibalmars/GraphRicciCurvature) is inversely proportional to the number of cores. In all cases we tested, the computation of Ricci flow is insignificant compared with the GPU training time, and can be handled easily even on a standard laptop machine. For example, it takes 18.9 s ± 73.5 ms on average to compute Ricci flow on the Cora graph with 2485 vertices and 5069 edges for 20 iterations on an Intel(R) Xeon(R) Gold 6140 CPU @ 2.30GHz 36 cores machine. On a 2018 Macbook Pro with 2.6 GHz 6-Core Intel Core i7 it takes 1min 6s ± 2.91 s on average.
>
> **[3] (Algorithm, graph sampling)**
>
> **Ans**: “I doubt that it’s possible to make any conclusive results about whether an edge is adversarial or not with 2-3 graphs.”
> Ans: All baseline algorithms (e.g., GCN, GAT) and ours use the same architecture of 2 layers, for a fair comparison. We add these missing experimental details in the revised paper and also share our anonymized codes at “https://anonymous.4open.science/r/43ab93f7-8b8b-4f76-a7c1-50ebc9bb0f6a/”. But we are not using only 2 graphs, because in the graph sampling phase we sample different graphs for each layer and each epoch. If our final run has 100 epochs, then we have 2*100=200 sampled graphs.
> The reviewer has a misunderstanding of graph sampling. Graph sampling is commonly used in the graph learning community to make the GNN scalable, such as GraphSAINT [Zeng et al. ICLR2020], LDS [Franceschi et al. ICML2019]. We randomly sample different graphs for every epoch. Please note **the sampling (Line 2 of Algorithm 1) is inside the training loop**. Different layers of the network use different sample graphs. The graph sampling technique does not require the sampled graphs to be similar to the original graph. To simply explain it, the final graph embedding is an expectation of the embeddings from all sample graphs. It has been fully understood and experimentally demonstrated in previous works. In our case,we use a probability matrix to sample the graph, which is similar to LDS.

---

> > ### Author Response · Authors · 2020-11-20
> > **We would like to point out that there seem to be quite a few misunderstandings, including Sinkhorn, graph sampling, defense baselines, etc.(2/2)**
> >
> > **[4](Design choices)**
> >
> > **Ans**: On the issue of graph density, we can control the density by tuning the parameters \sigma and \beta in our experiments. The number of edges in each re-sampled graph remains in the same order of the initial graph size.
> > On the issue of hyperparameters: for each of \sigma and \beta we tested five values. In our experiments we did not run parameter search on parameters \gamma, p, k. Below are the rationale for the choice of these parameters.
> > Sampling neighborhoods within 2 hops (k=2) is a very natural choice for many graph sampling methods, e.g. in GraphSAGE [Hamilton 2017]. Specifically in our case, we fix k=2 for two reasons: 1) due to the small world property of most random graphs and real world graphs the diameter is a small constant; 2) two nodes of three or more hops away are expected to have a long geodesic distance and thus the chance of being sampled is low. \gamma and p are parameters used in the computation of Ricci flow. In prior work [Ni 2019] the influence of these parameters on the computed curvature has been thoroughly evaluated. In short, the choice of different \gamma mainly introduces a global shift and scaling of the curvature values. The choice of p influences the convergence rate. In our experiments we fixed the value of \gamma and p as the suggested values as in [Ni 2019]. We also include a few new figures with different values of \gamma and p, which shows that there is no significant difference in a qualitative manner. Similarly, the choice of probability measure used in the curvature definition has been fully investigated in the prior work [Ni 2018] and [Ni 2019], which suggested to use the exponential distribution. This choice is also fixed in the experiments.
> >
> > **[5] (Datasets)**
> >
> > **Ans**: The three real-world datasets used in our paper are standard benchmark datasets in the evaluation of attack/defense schemes on GNNs, e.g., in the metattack paper (“Adversarial Attacks on Graph Neural Networks via Meta Learning), and a recent one (“Graph Structure Learning for Robust Graph Neural Networks”).
> >
> > **[6] (Baselines)** “Neither of GNN baselines (GCN, GAT, etc.) was designed for the defense of adversarial attacks, so choosing them for comparison is not fair. A comparison with previous works (indicated in “Adversarial attack on graphs.” in the related work section) is necessary. ”
> >
> > **Ans**: We do not agree with this evaluation. There are altogether 6 baselines in Table 1 & 2 of our original submission. Beside GCN and GAT, **the other 4 baselines (RGCN, GCN-Jaccard, GCN-SVD and Pro-GNN) are specifically designed for defense against adversarial attacks**. They were all listed in the related work section and described in greater details in supplemental material (Section C). The paragraph titled “Adversarial attack on graphs” **only discussed attack methods**. Defense methods were listed in the paragraph titled “Robustness of GNNs”. We don’t provide algorithms for randomly sampling edges because we have similar extensive experiments in ablation study (on hop count, see Table 3). Furthermore, per Reviewer2’s request, we added a new comparison with DropEdge as a baseline, which randomly drops edges.
> >
> > **[7] (Ablation)** ”Why is the performance of Ricci-GCN so different from GCN when there are 0 perturbations?”
> >
> > **Ans**: It seems this is a misunderstanding of our model. Our Ricci-GNN method restructures the original graph by using the Ricci flow metric. Therefore the training phase uses re-sampled graphs instead of the original graph, even when there is no edge perturbation.
> >
> > **[8] (Training)**
> >
> > **Ans**: Thanks for pointing this out. This is a typo in writing. For each method, 100 is the maximum training epoch number and we indeed used the best model based on validation performance. We clarified this point in the revised paper. We also added the code (anonymized) for your reference.
> >
> > At last, we kindly request the reviewer to re-evaluate our paper and reconsider the “confidence score”, considering the multiple misunderstanding points which are essential to our paper but actually well-known or basic in the GNN, or geometry research community (e.g. graph sampling, Sinkhorn).

---

> ### Author Response · Authors · 2020-11-23
> **Response to the Latest Comments**
>
> Thank you for timely response to our earlier post. Below we provide additional answers. Hope they address your latest concerns.
>
> **[1].** In simple words, the proposed approach uses some metric S (Ricci flow) that dictates how to sample graphs for training. The motivation for using Ricci flow is “that Ricci flow is a global process that tries to uncover the underlying metric space supported by the graph topology and thus embraces redundancy”. This claim cites previous papers, which in turn do not discuss what exactly is meant by “a global process that tries to uncover the underlying metric space”.
>
> **Ans:** Thanks for the suggestion. Essentially, Ricci flow is an iterative process where the weight of each edge (u, v) is adjusted proportional to its current curvature. As an outcome of this process, we obtain new edge weights such that curvature $\kappa$ is the same everywhere (and often zero). Upon an adversarial edge addition/removal, the disturbance to the distances between nodes in the graph is disseminated in a global manner; all pairs distances suffer similar (small) damage due to the adversarial attack. We added additional discussion to provide the intuition (see “robustness of Ricci flow metric” in Section 2.2).
>
>
> **[2].** Spectral embeddings also can be considered as a global metric, so some analysis on what properties of Ricci flow makes it more robust to attacks would be appreciated.
>
> **Ans:** Spectral embedding metric is well known to be sensitive to graph topology perturbation, whereas Ricci flow metric is robust to such perturbation (see the response in [#1]). In Figure 3, we demonstrated this difference. In the experiment section, our ablation study showed that sampling with spectral embedding metric has much less defense power than sampling with Ricci flow metric. Please see Table 3 for more details.
>
>
> **[3].** Also including random sampling in comparison would confirm that the effect is coming not from the fact that you use more graphs during the training, but from how you sample those graphs.
>
> **Ans:** To sample a graph, one needs a probability function defined on all node pairs. In Table 3, we have already provided two strong ablation models (HC and spectral). They sample the edges using hop count (shortest path) distance and spectral embedding distance respectively. By comparing these two metrics in the same random sampling framework, we showed the importance of Ricci flow metric. If you meant uniformly random sample edges, we have also provided an additional baseline, DropEdge, which randomly drops edges during each epoch. Ricci-GNN outperforms the baseline. Please refer to Appendix, Tables 5 and 6 for comparison.
>
>
> **[4].** In addition, as the paper is empirical and relies on the properties of Ricci flow which was discussed in previous works and was not addressed in the context of adversarial attacks, having more datasets (especially larger ones) in the experiments would improve the paper.
>
> **Ans:** The three datasets (Cora, Citesee and Polblogs) used in our paper covers the most commonly used datasets of state-of-the-art attacking and defense methods. See below for a list of these attacking/defense methods and their datasets.
>
> Attack methods:
>
> |Method|Reference|Datasets|
> |----|----|----|
> |Topo Attack-MinMax  | (Xu et al. 2019)                         | Cora, Citeseer                |
> |Meta-Attack               | (Zügner & Günnemann., 2019) | Cora, Citeseer, Polblogs|
>
>
> Defense methods:
>
> |Method|Reference|Datasets|
> |----|----|----|
> |RGCN             | ( Zhu et al.,2019)                            | Cora, Citeseer, PubMed|
> |GCN-Jaccard  | (Wu et al., 2019)                             | Cora, Citeseer, Polblogs|
> |GCN-SVD       | (Entezari et al., 2020)                     | Cora, Citeseer, Polblogs|
> |Pro-GNN         | (Jin et al., 2020b)                           | Cora, Citeseer, Polblogs, PubMed|
> |Topo-def          | (Miller et al., 2020)                         | Cora, Citeseer, Polblogs|
> |Cert-GN          | (Bojchevski & Günnemann.,2019)  | Cora, Citeseer               |
>
> To extend the experiments to larger graphs, the computational bottleneck is the attacking methods, not our method. SOTA attacking methods (Meta-Attack and MinMax-Attack, both used in our paper) compute gradients on the full adjacency matrix, and thus cannot be applied to larger graphs. Our algorithm scales well to large graphs. The computation of Ricci flow metric is dominated by the computation of Ricci curvature, which can be efficiently computed on large scale graphs with up to 30K nodes and 300K edges, as shown in previous work (Ye et al. ICLR 2020).

---

### Official Review · AnonReviewer3 · 2020-10-29

**Rating:** 5
**Confidence:** 5

**Review:**

##########################################################################

Summary:
The paper proses a new adversarial (poisoning) defense based on a known graph reweighting scheme known as the ricci curvature. The ricci curvature assigns a weight to each edge that captures the graph structure, i.e. the value reflects whether the edge is an inter-community connection or an intracommunity connection. Empirically, the ricci curvature is known to be more robust w.r.t. random edge insertions/deletions. The authors propose a new sampling method based on the ricci curvature and use it within their novel training scheme. Empirically, the effectiveness of their approach is shown via experiments on synthetic SBM graphs. Moreover, the authors use a random attack and Metattack on various datasets. They show superior performance to multiple baseline architectures/defenses.

##########################################################################

Reasons for rating:

Overall it is an interesting work and the empirical performance seems to be good. However, neglecting the very weak random attack and the experiment on synthetic data, the authors effectively only evaluate against one strong attack. Hence, the question arises if the defense is solely effective against the characteristic of Metattack? I would recommend to add at least one further strong attack.

Furthermore, it is not clear if a transfer attack is used (a surrogate used for Metattack). It would be very interesting to see if the ricci curvature calculation itself is adversarially robust. In the chosen setup this fact is obfuscated.

Last, the authors cite but do not compare to the Curvature Graph Network which also uses ricci curvature instead of the widely used symmetric normalization of the adjacency matrix (e.g. as a GCN). Hence, it is not clear to the reader if the sampling scheme/training scheme or the ricci curvature is the major reason for adversarial robustness.

##########################################################################

Pros:
+ Interesting and promising approach
+ Consistently improved performance over the baselines
+ Interesting analysis of their defense via SBM graphs


Cons:
- Only one strong attack on real-world graphs is used to benchmark to other architectures.
- Is the ricci curvature itself robust w.r.t. adversarial attacks? The authors seem to use transfer attacks and the referenced literature claims only robustness against random attacks and this is also only evaluated empirically.
- Curvature Graph Network should be added as a baseline.
- The authors do not discuss the space and time complexity. Only the time cost for the ricci curvature is discussed. Moreover, the authors use the two-hop neighborhood for adding potential edges — this can still be very expensive specifically for power-law graphs. A discussion would be appreciated.

Further points:
- The proposed sampling based training scheme seems to be highly related to adversarial training. The authors should add a corresponding discussion.
- The paper lacks clarity at some points and has inconsistencies in notation. For example, in Section 2 "F" is not introduced. S denotes the geodesic distance (aka length of shortest path) which is denoted by d(x, y) in Section A.
- At some points the authors say that a graph is sampled in each "iteration" (epoch) and sometimes for every layer.
- Figure 2: What is H_0, H_1, L?
- Section 1.1: The authors should make clear that Figure 3 is an example and does not imply superior robustness in general.
- What are the limits of ricci curvature? Beyond some level of perturbation, there should be a tipping point (i.e. communities cannot be distinguished anymore).
- Related work: There are many more (relevant) attacks/defenses. Please add them and/or make clear that your discussion is not exhaustive.

---

> ### Author Response · Authors · 2020-11-20
> **Added extra experiments (new attack, new baseline), and made some clarifications.**
>
> Thanks for the comments. We address the main concerns one-by-one.
>
> Cons:
>
> **[1](Only one strong attack)**
>
> **Ans**: Per your request, we added a new attack called Topological attack-MinMax from the paper: “Topology Attack and Defense for Graph Neural Networks: An Optimization Perspective”. The result has been added to the appendix (Table 5). Our Ricci-GNN achieves the best performance on all datasets under this new attack.
>
> **[2] (Is the ricci curvature itself robust)**
>
> **Ans**: No. Ricci curvature itself is not robust. Ricci curvature, defined on an edge, can be considered as a measure of how much one edge changes the network connectivity. Edges of negative curvature (e.g., a critical edge connecting two communities), if deleted, may completely change the global network layout. Ricci flow (which is used in this paper) is to modify the edge weights such that all edges have the same curvature. The new weights are observed in [Ni 2018] to be robust to *random* edge insertion/deletion. Inspired by this observation, we hypothesized that Ricci flow metric can improve robustness against *adversarial attacks*. This is validated empirically.  We added additional discussion to provide the intuition (see “robustness of Ricci flow metric” in Section 2.2).
>
> **[3](Curvature Graph Network)**
>
> **Ans**: Thanks for the suggestion. We added curvature graph network (CurvGN) as a baseline method. The design of CurvGN is to use curvature as additional structural information in the training. It does not consider robustness and from our experiments there does not seem to be extra benefits of robustness against adversarial attacks. The new results are shown in our Appendix B 3.2.
>
> **[4](The authors do not discuss the space and time complexity)**
>
> **Ans**: The time complexity of computing Ricci curvature on an edge using Sinkhorn distances is O^tilde (d^2) where d is the maximum degree. So running one Ricci flow iteration requires O^tilde(|E|*d^2). Ricci flow is only computed once at the beginning of the GNN algorithm. The space complexity is O(|V|^2).
>
> Further points:
>
> **[1](The proposed sampling…)**
>
> **Ans**: The adversarial training method is trying to introduce adversarial examples and increase the model’s robustness with regularization terms that neutralize the effect of the introduced adversarial samples. However, our method directly changes the graph’s structure to improve the model’s robustness.
>
> **[2](The paper lacks...)**
>
> **Ans**: Thanks for pointing this out. We have updated the paper accordingly. Recall that in Section A.2, we use Ricci flow (Eq A.3 same as Eq 2.3) to iteratively update the edge $w^{(t+1)}(x,y)$. So F is the edge weight matrix for this resultant weight after convergence: $F_{ij} = w^*(i,j)$. And S is the matrix for the geodesic distance by running the all-pair shortest paths on weights in $F$. Therefore $F(x,y) = 0$ if there is no edge $xy$. But $S(x,y)$ is non-zero as long as there is a path connecting x and y.
>
> **[3](At some points…)**
>
> **Ans**: For each epoch, for each layer, we sample a new graph. Therefore, if we run for 100 epochs on a GNN with 2 layers, we sample $2\times 100$ graphs in total.
>
> **[4](Figure 2...)**
>
> **Ans**: H_0 is the initial feature matrix (H_0 = H in Sec 2.1) with dimension |V| (number of nodes) * |D| (feature dimension). H_1 is the transformed feature after applying the first layer of GNN on H_0. L is the final output of GNN. If it is a binary node classification task, L is of dimension |V| (predicted score for each node). It is explained in the third paragraph of section 2.1. We also change the subscript of H into a superscript to make it consistent with the context.
>
> **[5](Section 1.1...)**
>
> **Ans**: It’s correct that Figure 3 serves as a motivating example to visually compare different graph metrics. It is observed in the literature (such as Ni et al., 2015; Sandhu et al., 2015) that Ricci curvature is more robust for several real-world graph datasets and is a useful prior for many clustering/classification tasks on such datasets.
>
> **[6](What are the limits…)**
>
> **Ans**: We agree that when the perturbation rate increases, eventually all learning methods fail as the input graph connectivity becomes completely different from the ground truth. In our experiments (and also in prior literature) the highest perturbation rate we tested is 25% -- which is already significantly high for any practical settings. It seems that 25% perturbation rate is the tipping point for all other defense methods (for Polblogs), while our method can still provide reasonable results.
>
> **[7](Related work…)**
>
> **Ans**: Thank you! We edited our related work as suggested.

---

> > ### Comment · AnonReviewer3 · 2020-11-24
> > **Response**
> >
> > Thank you for your feedback. Adding the „Topological attack-MinMax“ makes the empirical evaluation much more convincing! The two major remaining concerns are:
> > - I am still not fully convinced with the discussion about the robustness of the ricci curvature itself.
> > - The randomized training scheme is somewhat ad hoc and no convincing argument is given why it was chosen or how it relates to robust training
> >
> > I have increased my score to 5 with a strong tendency towards 6.

---

> > > ### Author Response · Authors · 2020-11-25
> > > **Robustness and Randomized Training**
> > >
> > > Thank you for the positive feedback. Below are the answers to your latest two questions. Hope they can further increase your confidence on our paper.
> > >
> > >
> > > **Q1.**  I am still not fully convinced with the discussion about the robustness of the ricci curvature itself.
> > >
> > > **Ans:** Ricci curvature and Ricci flow are conceptually different. Our paper is based on the robustness of Ricci flow metric, not Ricci curvature. The Ricci curvature of an edge $(u, v)$ measures the connectivity in the neighborhood of $u$ and $v$ in the graph. Using curvature as a feature in GNN training helps to improve accuracy. But it is still local information and does not improve the robustness. As shown in the additional results we added per your suggestion (Tables 5 and 6), CurvGN is not a strong defense baseline.
> > >
> > > On the other hand, Ricci flow metric encodes the graph topology in a global manner. It uses a global iterative PDE-like process to change edge weights, so that all edges have the same curvature $\kappa$. To further explain the robustness of Ricci flow, let us first consider the original continuous setting of Ricci flow on a closed surface, the converged curvature $\kappa$ is determined by the global topology (e.g., whether the surface is topologically a torus or a sphere) and independent of the initial geometry. This property seems to have carried over to the discrete setting on graphs. That is, unless one adds/removes a significant fraction of edges (up to the point that the global graph structure is significantly altered), the Ricci flow process converges to a similar global curvature $\kappa$ and produces a similar Ricci flow metric.
> > >
> > > In practice, we observe that the impact of an addition/removal of edges is diffused in the graph through the Ricci flow process. In turn, **the Ricci flow distance between any pair of nodes will not be significantly changed when we perturb a limited set of edges, even adversarially.** This is illustrated in Figure 3, in which we show that Ricci flow metric is robust against edge perturbation, whereas spectral distance is not.
> > >
> > > Please also refer to “robustness of Ricci flow metric” in Section 2.2 for more discussions.
> > >
> > >
> > > **Q2.** The randomized training scheme is somewhat ad hoc and no convincing argument is given why it was chosen or how it relates to robust training
> > >
> > > **Ans:** The random sampling of graphs is a natural choice when we rely on the Ricci flow metric instead of the original graph. Each node samples edges to other nodes with probability taken from a Gaussian kernel. This produces probabilistic graph realizations on which we can train the GNN.
> > >
> > > Other alternatives will not fit: If we take a complete graph using the Ricci flow metric as the adjacency/distance matrix, GNN training can be expensive and does not scale. If we use a hard threshold (or any other fixed method) to select edges, we end up with a fixed graph that may lead to overfitting, as a single fixed subgraph graph is a biased approximation to the Ricci flow metric.
> > >
> > > Indeed, the idea of random sampling graphs from a distribution has been a popular choice in recent GNN studies. Examples include GraphSAINT [Zeng et al. ICLR2020], LDS [Franceschi et al. ICML2019], etc. We note that random sampling is not the main reason for the GNN robustness. We have shown in ablation studies that the same sampling approach with other underlying metrics (spectral and hop count) are not as effective (Tables 3). The robustness of our method is mainly inherited from the Ricci flow metric, which was explained in the previous question.

---

### Author Response · Authors · 2020-11-20
**Summary of Rebuttal**

We thank all reviewers for their feedback and suggestions. Here we summarize the major clarification and new experiments per their suggestions.

[1]. We added more references to related work.

[2]. We added two extra baselines as the reviewer requested to appendix: Curvature Graph Network and DropEdge.

[3]. We added One SOTA attack method to appendix: Topological attack-MinMax.

[4]. We clarified details of the experimental setting. During attacking phase, we follow the setting of (Zugner and Gunnemann ICLR 2019). We run the meta-attack method gradient decent for 100 iterations to generate the attacked graph. During the defense phase, we follow the setting of Pro-GNN: we run the training for 100 epochs, and select the model with the best performance on the validation set.

[5]. We added accuracy for different methods on 0 perturbation rate in Table 3.

[6]. We added more experimental details about the number of layers of all GNN baselines.

---

### Decision · Program_Chairs · 2021-01-07
**Final Decision**

**Decision:**

Reject

**Comment:**

The paper proposes a new defense against adversarial attacks on graphs using a reweighting scheme based on Ricci-flow. Reviewers highlighted that the paper introduces interesting ideas and that the use of Ricci-curvature/flow is a novel and promising contribution. Reviewers also recognized that the paper has significantly improved after rebuttal and clarified some aspects of their initial reviews.

However, there exist still concerns around the current version of the manuscript. In particular, important aspects of the method and algorithm, as well as some design choices are currently unclear. This includes evaluating and discussing robustness, training method, and practicality/improvements in real-world scenarios. I agree with the majority of the reviewers that the current version requires an additional revision to iron out the aforementioned issues. However, I also agree with the reviewers that the overall idea is promising and I'd encourage the authors to revise and resubmit their work with considering the feedback from this round of reviews.